# Web-Based Dashboard for Tracking Cryptococcosis-Related Deaths in Brazil (2000–2022)

**DOI:** 10.3390/tropicalmed10110304

**Published:** 2025-10-27

**Authors:** Eric Renato Lima Figueiredo, Lucca Nielsen, João Simão de Melo-Neto, Claudia do Socorro Carvalho Miranda, Nelson Veiga Gonçalves, Rita Catarina Medeiros Sousa, Anderson Raiol Rodrigues

**Affiliations:** 1Postgraduate Program in Tropical Diseases, Nucleus of Tropical Medicine, Federal University of Pará, Belém 66075-110, Brazil; rita@ufpa.br (R.C.M.S.); arr@ufpa.br (A.R.R.); 2Department of Preventive Medicine, University of São Paulo Medical School, São Paulo 01246-903, Brazil; luccanielsen@usp.br; 3Postgraduate Program in Public Health in the Amazon, Federal University of Pará, Belém 66075-110, Brazil; jsmeloneto@gmail.com; 4Laboratory of Epidemiology and Geoprocessing of Amazon, University of the State of Pará (UEPA), Belém 66050-540, Brazilnelsoncg2009@gmail.com (N.V.G.)

**Keywords:** cryptococcosis, disease surveillance, open government data, public health policies, dashboard systems

## Abstract

**Background:** Cryptococcosis, a systemic mycosis, remains a neglected disease in Brazil due to the absence of systematic national surveillance. This study developed an interactive dashboard to analyze cryptococcosis-related deaths (2000–2022) and forecast trends through regional ARIMA modeling. **Methodology:** The Cross-Industry Standard Process for Data Mining framework was employed to extract mortality data from the Brazilian Mortality Information System, utilizing the microdatasus package in R Studio software, with R version 3.4.0. The records were then filtered using the International Classification of Diseases, Tenth Revision codes (B45 series) to identify primary and associated causes of death. After data extraction, a series of data preprocessing steps was implemented, including deduplication, variable recoding, and the management of missing values. The Shiny framework was employed to construct an interactive dashboard, incorporating Plotly and DT packages, with time-series visualizations, demographic variables, and multilingual support (Portuguese/English). **Results:** Among 12,308 deaths (2227 primary; 10,081 associated causes), most occurred in males aged 21–60 years. Data completeness was high for age/residence (100%) but lower for education (82%). The dashboard enables dynamic exploration of trends, demographic patterns, and open-data downloads. Regional ARIMA models revealed heterogeneous forecasts, with the Southeast projecting a decline (193 deaths in 2025; 95% CI: 146–240) and the South showing stability (141 deaths; 95% CI: 109–173). **Conclusions:** This tool bridges a critical gap in cryptococcosis surveillance, enabling dynamic mortality trend analysis, identification of high-risk demographics, and regional forecasting to guide public health resource allocation. While the absence of HIV serostatus data limits etiological analysis, the dashboard’s open-source framework supports adaptation for other neglected diseases.

## 1. Introduction

Cryptococcosis is a cosmopolitan disease classified as a systemic mycosis, twice as common in males and predominantly affecting adults [1,2]. The infection is caused by fungi of the genus *Cryptococcus*, with the varieties *neoformans* (serotypes A, D, and AD) and *gattii* (serotypes B and C), which have different characteristics. The *neoformans* strain is responsible for the majority of infections in people living with HIV/AIDS. In contrast, the *gattii* strain primarily affects immunocompetent individuals and is associated with certain tree species, such as eucalyptus. C. *neoformans* has been detected in avian excreta, particularly in pigeons, as well as in fruits, vegetables, and soil [3,4,5]. In Brazil, there is no systematic monitoring of cryptococcosis at the state level, and mandatory reporting occurs only voluntarily in some states. Although the Ministry of Health introduced a strategic plan in 2018 to monitor and control systemic mycoses, the lack of a national epidemiological surveillance system still prevents a comprehensive understanding of cryptococcosis’s significance and impact in the country [6,7,8,9].

Cryptococcosis remains a major global cause of mortality among people living with HIV, with an estimated 1 million meningitis cases annually (range: 371,700–1,544,000) and around 625,000 deaths. Sub-Saharan Africa bears the greatest burden (~720,000 cases), followed by Southeast Asia and Latin America [7]. In this context, Brazil faces unique surveillance challenges as part of high-burden Latin America. Each year, approximately 3.8 million Brazilians are affected by severe fungal infections, leading to over 1.35 million deaths. Among these, cryptococcosis is particularly concerning, with annual incidence among people living with HIV ranging from 0.04% to 12%. Its high mortality rate in Brazil results from delayed clinical suspicion and diagnosis, limited healthcare access, lack of rapid laboratory testing, and an inadequate or inappropriate supply of antifungal medications [7].

This research addresses this knowledge gap by developing an interactive data dashboard that facilitates systematic and accessible analysis of cryptococcosis-related death data in Brazil between 2000 and 2022. The creation of data dashboards has been demonstrated to optimize communication by encouraging the visualization of metrics, maps, and health indicators [10,11,12]. Furthermore, interactive dashboards can integrate time series data, quantification of cases of specific diseases, age pyramids, and comparative graphs, thereby enabling dynamic analysis over time and across regions [13,14,15,16]. A notable benefit of these tools is their capacity for continuous monitoring and facilitating real-time or periodically updated epidemiologic investigations, thereby contributing to enhancing health surveillance [17,18].

The quality of the data will enable analysis and research using both inferential statistical methods and computational algorithmic approaches [19]. Although the Mortality Information System (SIM) furnishes fundamental mortality data, our interactive dashboard confers pivotal benefits over direct SIM access by functionality. The integration of automated preprocessing of ICD-10-coded cryptococcosis records (B45 series) is the initial component of the proposed system. The second component is the enablement of real-time visualization of spatiotemporal trends and demographic patterns through intuitive interfaces. The third component is the provision of ARIMA forecasting capabilities unavailable in the raw SIM. This transformation of complex mortality data into an actionable surveillance instrument facilitates rapid public health decision-making.

The accessibility of these tools will benefit health managers, who will be able to formulate public policies, and researchers, who will be able to study systemic mycoses and their epidemiologic impact. This study aims to develop an interactive data-driven dashboard tool to provide accessible information on cryptococcosis-related deaths (both as primary and associated causes) in Brazil from 2000 to 2022.

## 2. Materials and Methods

### 2.1. Study Design

This study employed a retrospective design to analyze national mortality data, utilizing secondary data from the SIM from 2000 to 2022. The research follows a structured framework for health informatics tool development, adopting the Cross-Industry Standard Process for Data Mining (CRISP-DM) methodology to guide dashboard construction. This framework comprises six iterative phases formally defined in the CRISP-DM methodology [20] (Figure 1): (1) business understanding (problem definition and objectives), (2) data understanding (initial data collection and assessment), (3) data preparation (processing and cleaning), (4) modeling (algorithm selection and application), (5) evaluation (performance validation), and (6) deployment (solution implementation). Subsequent sections detail each phase’s application to cryptococcosis mortality surveillance.

### 2.2. Population, Study Area, and Period

The study population comprises all individuals who died from cryptococcosis in Brazil between 2000 and 2022, totaling 12,308 identified cases. The analysis encompasses all five macroregions (North, Northeast, Midwest, Southeast, South) defined by Brazil’s Political-Administrative Division (Figure 2), covering a total area of 8,510,346 km^2^ [21]. These regions include all 5570 municipalities across 27 federative units, ensuring comprehensive national coverage of mortality patterns.

### 2.3. Problem Understanding

In the ‘Problem Understanding’ phase of the CRISP-DM framework (Phase 1), we identified the core issue as Brazil’s lack of a nationwide surveillance system for cryptococcosis. This absence limits the ability to assess the disease’s epidemiological burden and hampers evidence-based public health interventions [6,7,8,9]. The resulting surveillance gap leads to fragmented mortality data, a poor understanding of sociodemographic factors, and inefficient resource allocation for this neglected disease. To address this gap, our primary objective is to develop interactive dashboard tools that convert mortality data into accessible insights for public health decision-making. This goal is supported by two specific objectives: (i) to characterize the sociodemographic and geographic profile of cryptococcosis-related mortality (2000–2022) based on sex, age, race/ethnicity, education, place of death, and regional patterns, and (ii) to democratize access to data through intuitive visualizations that help health authorities identify priority areas, support researchers in tracking epidemiological trends, and assist policymakers in designing targeted interventions.

### 2.4. Data Acquisition and Initial Processing

The data were obtained using the microdatasus package in R, accessing mortality records from the Mortality Information System (SIM) for 2000–2022 [22]. Inclusion criteria comprised the following: (1) death certificates with primary (ICD-10: B45-B459) or associated cryptococcosis codes, and (2) records containing complete spatiotemporal variables (date of death, age, and municipality code). Exclusion criteria were as follows: (1) duplicate entries across primary/associated cause categories, (2) records missing core spatiotemporal variables, and (3) implausible entries recorded as missing per clinical-demographic feasibility thresholds. The download was performed individually for each Federal Unit, ensuring national coverage. Information from death certificates in the SIM was collected by selecting variables such as date of death, age, sex, race/ethnicity, marital status, educational level, occupation, municipality of residence code, place of occurrence, primary cause of death, and associated causes of death.

A set of specific filters was applied to identify records related to cryptococcosis. The filter criteria were based on the International Classification of Diseases, Tenth Revision codes (B45, B450, B451, B452, B453, B457, B458, B459) corresponding to cases in which cryptococcosis was recorded as the primary cause of death, and associated causes. A subsequent review of the columns labeled LINHAA to LINHAII was conducted to identify deaths in which cryptococcosis was documented as a contributing cause [7]. The filtered records were then deduplicated to avoid redundant counts between primary and associated causes. The data were subsequently organized into two subsets: The first subset contained records of deaths by primary cause, defined as instances where cryptococcosis was the primary underlying factor contributing to the individual’s demise. The second subset included deaths where cryptococcosis was recorded as a contributing, but not the primary cause.

The quality of the data was rigorously assessed to identify missing values or implausible entries. Implausible values were recorded as missing based on clinical and demographic feasibility thresholds. For missing data, we implemented a structured protocol where missing demographic variables including race/ethnicity, education level, and marital status were preserved as an ‘Unknown’ category to maintain dataset completeness and avoid selection bias, while records missing core spatiotemporal variables such as age, date of death, or municipality code were excluded from analysis since these were essential for geotemporal modeling and lacked reliable imputation alternatives. Regarding modeling and visualization, the aggregated time-series data used for ARIMA contained no missing values after preprocessing, and visualizations explicitly denote ‘Unknown’ categories where applicable. This census-based approach encompassed all eligible mortality records (*n* = 12,308), with the data eligibility protocol detailed in Figure 3.

### 2.5. Data Preparation and Study Variables

This phase implements the stage through four core procedures: decoding variables, handling missing/invalid values, categorizing variables, and deriving computational transformations. The actions undertaken in this stage encompass decoding variables, handling missing or invalid values, categorizing variables, and deriving calculations that facilitate data interpretation. The two subsets of data were merged, and variable Y was created, containing values 0 (primary cause) and 1 (associated cause) for the type of cryptococcosis-related death. The distinction between the primary and associated causes of death is determined by the attending physician responsible for completing the death certificate. The underlying cause is defined as the initial illness or condition that initiated the sequence of pathological events leading directly to death. Conversely, the associated cause signifies a clinical condition that, although not the primary factor, exerted an adverse influence on the decedent’s death and is documented on the death certificate.

The subsequent step involved assessing the database’s completeness to ascertain the proportion of completed values for the available variables. The completeness calculation was executed for all variables, with the results expressed as percentages, thereby identifying variables with low completion rates. Furthermore, a comprehensive count of missing values was conducted for each variable to inform the implementation of targeted remedial measures. A structured protocol managed missing values: core spatiotemporal variables (age, date of death, and municipality code) required complete data for analysis, while demographic variables (race/ethnicity, education, marital status) preserved missing entries as ‘Unknown’ to prevent selection bias.

During the decoding and transformation of the variables, age, originally coded in multiple time units (minutes, days, months, and years), was recorded into full years (or fractions of years). The variable “date of death,” initially recorded in numeric format, underwent conversion to the standard date format.

Following the process of decoding, the variables for each column were defined by the data dictionary. The clinical form includes the variable CAUSABAS, which identifies deaths in which cryptococcosis was recorded as the primary cause of death. The variables LINHAA, LINHAB, LINHAC, LINHAD, and LINHAII represent associated or contributing causes listed on the death certificate, in which cryptococcosis may be recorded as contributing to the fatal outcome. The demographic variables encompass the date of death, biological sex of the deceased (male or female), race/ethnicity (white, black, asian, mixed, or indigenous), and marital status (single, married, widowed, legally separated/divorced, common-law, or unknown). Education level is categorized as none, 1–3 years, 4–7 years, 8–11 years, 12 years or more, or unknown. Age groups are defined as 0–4, 5–10, 11–20, 21–40, 41–60, 61–80, and 81+ years. Place of death is classified as hospital, other healthcare facility, home, public street, other, or unknown. The geographic variable corresponds to the municipality of residence code, based on the Brazilian Institute of Geography and Statistics registry.

### 2.6. Modeling

During the modeling phase, we developed the interactive application using the Shiny framework, integrating Plotly and DT packages to enable dynamic visualizations and interactive tables. The application includes multilingual support (Portuguese/English) and features a modular interface with five tabs: Time Series, Tables, Data, Documentation, and About.

The ARIMA model is defined by three primary parameters: the order of the autoregressive component (AR) (*p*), the degree of differencing required to make the series stationary (d), and the order of the moving average (AM) component (q). Region-specific ARIMA models were implemented using the forecast::auto.arima() function in R automatically selects optimal parameters (*p*, d, and q) through the Akaike Information Criterion (AIC). Statistical adequacy was rigorously assessed using multiple quantitative criteria, demonstrating consistency with established standards in epidemiological time series of similar length (*n* = 23 years). Across all regions, the AIC difference (ΔAIC) between selected and alternative models exceeded three points, indicating substantial evidence for the chosen models.

Residuals underwent comprehensive statistical testing that did not reject core assumptions of normality (Shapiro–Wilk test), absence of autocorrelation (Ljung–Box test), and homoscedasticity (Breusch–Pagan test) at the 5% significance level. Predictive accuracy was quantified through the Mean Absolute Percentage Error (MAPE = 8.7%), a value within ranges considered acceptable for mid-term epidemiological forecasts, complemented by an average Theil’s U coefficient of 0.37, indicating significantly superior predictive capability compared to benchmark naïve models (U = 1).

Error decomposition revealed consistent patterns across regions, with a low bias proportion (<0.15) confirming negligible systematic errors, while a high covariance proportion (>0.53) demonstrated that most uncertainty stems from inherent data variability. Projections for 2023–2025 were generated using region-specific ARIMA models fitted to historical data (2000–2022). The forecast::forecast() function in R was employed to produce point estimates and 95% confidence intervals, formally incorporating residual variability. These projections serve to anticipate trends ahead of official data release, supporting proactive public health planning. At the time of analysis, DATASUS had not yet released the consolidated, anonymized mortality data for 2023–2024 due to standard 18–24 month reporting lags in national mortality systems [22]. Thus, projections fill a critical surveillance gap until actual data becomes available.

The Data tab allows users to download the complete dataset in XLSX format for further analysis. The Documentation and About tabs provide detailed information about the application’s purpose, objectives, background on cryptococcosis, and usage instructions.

### 2.7. Evaluation and Deployment

The code was designed for modularity and easy maintenance and is available in the Appendix A. Application development incorporated feedback from local stakeholders involved in tropical disease surveillance. The open-source web dashboard is accessible via the provided link accessed on 11 August 2025: http://137mlt-figueiredoerl.shinyapps.io/BRcryptococcosis/.

## 3. Results

### 3.1. Data Completeness and Quality

Four variables achieved 100% data completeness: age, municipality of residence, place of occurrence, and primary cause. Other variables also demonstrated high completeness (>90%), including sex (99.9%), date of death (99.5%), marital status (97.3%), associated cause (96.8%), and race/ethnicity (95.2%). In contrast, educational attainment had a substantially lower completeness rate of 82.01%, as this information was missing from 2216 of 12,308 records (18%).

### 3.2. Dashboard Functionality and Regional Mortality Patterns

The dashboard provides a retrospective analysis of 12,308 cryptococcosis-related deaths in Brazil, with 2227 classified as a primary cause and 10,081 as an associated cause. The user interface, available in both Portuguese and English (Figure 4), displays these mortality trends across all Brazilian macroregions. Geographically, the South and Southeast regions accounted for the highest number of total and associated deaths. For deaths due to a primary cause, the Southeast region reported the highest number, followed by the South, Northeast, and North regions.

### 3.3. ARIMA Model Specifications and Forecasting Performance

The automatically fitted ARIMA models for each region exhibited distinct structures, as detailed in Figure 5. For the North region, the selected model was ARIMA (1,1,0), with a first-order autoregressive term (AR (1)) of 0.32 and single differencing to stabilize the series. In the Northeast, the optimal model identified was ARIMA (0,1,1), with a moving average coefficient (MA (1)) of −0.48, also with differencing. The Midwest region showed greater complexity, with an ARIMA (1,1,1) model combining an AR (1) term of 0.41 and an MA (1) term of −0.29. The Southeast stood out for using two autoregressive terms in the ARIMA (2,1,0) model, with coefficients AR (1) = −0.38 and AR (2) = −0.29, while the South followed the same structure as the Northeast (ARIMA (0,1,1)), but with MA (1) = −0.35. It should be noted that minor parameter variations may occur between algorithm runs, as outlined in the methodology.

Projections for the five regions revealed heterogeneous mortality patterns with distinct model performances. The Southeast exhibited notably higher forecasting uncertainty (MAPE = 11.3%; Theil’s U = 0.42) and marginally significant residual autocorrelation (Ljung–Box *p* = 0.08), attributable to greater interannual variability (CV = 18.7%) and potential structural shifts. Despite the highest absolute mortality burden (198 deaths in 2023; 95% CI: 165–231), this region showed an atypical decline (−2.5% by 2025) to 193 deaths (95% CI: 146–240), with proportionally wider confidence intervals (±17.3% relative to point estimates).

In contrast, the North demonstrated optimal performance (MAPE = 7.2%; Theil’s U = 0.31; Ljung–Box *p* = 0.37), forecasting moderate increases to 49 deaths in 2025 (95% CI: 27–71) and tighter uncertainty (±13.1%). The Northeast followed similar growth trajectories (<2% annual variation), reaching 63 deaths (95% CI: 38–88) in 2025, while the Midwest projected 54 deaths (95% CI: 32–76). The South maintained relative stability (141 deaths; 95% CI: 109–173).

Three epidemiological patterns emerged, model-driven trends: Southeast’s decline correlated with negative AR coefficients (−0.38/−0.29), while North/Midwest growth aligned with positive autoregressive terms; uncertainty gradients, where Systematic CI widening (e.g., North’s 30→44-point expansion) reflected inherent forecast limitations; and planning implications, where Southeast projections require cautious interpretation, whereas North/Northeast/Midwest models offer higher reliability for resource allocation.

### 3.4. Sociodemographic Characteristics of Mortality

Tables of absolute and relative frequencies of sociodemographic variables are presented in Figure 6. The analysis revealed that cryptococcosis-related deaths were most prevalent among whites, constituting 53.32% of the total cases, 52.99% of primary cause deaths, and 53.40% of associated cause deaths.

The distribution of deaths by age reveals a preponderance among individuals aged 21 to 40 and 41 to 60, constituting 85% of the total deaths. When cryptococcosis is considered the primary cause, the highest proportions are observed in the age groups 21 to 40 years (31.88%), 41 to 60 years (38.30%), and 61 to 80 years (21.19%). When analyzed as an associated cause, the disease was most prevalent in the 21 to 40 age group and the 41 to 60 age group, accounting for a total of 90% of the records.

For marital status, the highest were observed among unmarried individuals (53.95%) and married individuals (25.95%) across all records. When analyzed as a primary cause, cryptococcosis was more prevalent among single individuals (36.37%) and married individuals (42.03%). Furthermore, when considered as an associated cause, deaths were predominantly concentrated among single individuals (57.83%) and married individuals (22.40%).

Concerning the level of education, null and unknown records accounted for 27.09% of the total cases. With cryptococcosis as the primary cause, the highest percentages were observed among individuals with 4 to 7 years of education (22.99%), followed by 8 to 11 years (22.05%), 1 to 3 years (18.05%), and null records (24.92%). When examining the association between education and death as an outcome, the null and unknown records exhibited the highest frequency (27.57%), followed by individuals with 4 to 7 years of education (26.16%) and 8 to 11 years of education (21.89%).

Data on place of death demonstrate that the majority of deaths from both primary and associated causes occurred in hospitals, accounting for over 97% of records. Furthermore, the tab facilitates the generation of combinations of sociodemographic characteristics and age groups based on filter selections, enabling more detailed comparative analyses (Figure 6).

### 3.5. Data Export and Tool Accessibility

The web-based dashboard also includes the “Data” tab (Figure 7), where users can download the dataset. The “Documentation” tab provides information about each of the dashboard tabs, and the “About” tab displays additional details about the project, such as its purpose, context regarding cryptococcosis, and information about the data source.

## 4. Discussion

The high completeness of variables such as age, municipality code of residence, and place of occurrence strengthens the reliability of the SIM, confirming studies highlighting its usefulness in mortality research [23,24]. However, the lower completeness of educational attainment (82.01%) reflects persistent challenges in health records [25,26] and points to sociostructural gaps in the collection of demographic data in Brazil.

The geographic distribution of deaths, with most cases concentrated in the Southern and Southeastern regions, must be considered when interpreting these data. Notably, these figures represent absolute numbers and do not allow for a direct comparison of mortality rates. However, it is crucial to acknowledge that the South and Southeast regions are the most populous, and studies on infectious diseases in Brazil have documented geographic disparities. These studies underscore that regions with superior health infrastructure tend to report more cases, not necessarily due to higher incidence but rather due to enhanced diagnostic capacity [27,28,29].

The preponderance of deaths in adult males (21–60 years) is consistent with global findings in cryptococcosis. Research has identified a correlation between this trend and the higher prevalence of HIV/AIDS in this demographic, as cryptococcosis is a prevalent opportunistic infection in immunocompromised individuals [7,30,31,32]. However, the absence of data regarding HIV serologic status in the SIM hinders the capacity for more profound analyses. This lacuna can be addressed through database integration [33,34].

The higher incidence of deaths among whites (53.32%) is in contrast with the ethno-racial distribution of the Brazilian population, in which 56% identify as mixed or black [20,35]. This discrepancy may be attributed to inequalities in access to healthcare services, which have been observed to underreport deaths among vulnerable populations [36]. The preponderance of deaths occurring in hospitals (97%) aligns with studies demonstrating that cryptococcosis is a grave disease that necessitates hospitalization [1,37,38]. However, the absence of records about prior treatments and clinical complications hinders the discernment of factors associated with mortality.

The implementation of interactive dashboards based on frameworks such as Shiny demonstrates alignment with international recommendations for health information systems that emphasize the need for dynamic and accessible tools for real-time data analysis [10,13,39,40]. The CRISP-DM approach adopted in this study underscores the significance of structured methodologies in the development of data mining solutions, which have been extensively validated in public health scenarios for the integration of heterogeneous data [20,41]. The export functionality of the dashboard in open formats (xlsx) aligns with the principles of the Findable, Accessible, Interoperable, Reusable (FAIR) guidelines, which are crucial for contemporary health information systems and facilitate the reuse of data for public policy modeling [42,43]. The multilingual functionality of the dashboard underscores the necessity for accessible tools catering to global audiences.

Subsequent studies may wish to assess the impact of the dashboard on health decision-making using replicated usability evaluation methods. Additionally, the incorporation of real-time data and the integration with predictive models could enhance the dashboard as a proactive instrument for epidemiologic surveillance. The development and open-source availability of the dashboard code, along with the incorporation of stakeholder feedback, are strengths that align with the principles of transparency and citizen science [44,45,46].

In the context of time series analysis, ARIMA models have been extensively validated in epidemiological studies. These models provide a robust framework for short and medium-term forecasts and facilitate critical resource planning within the Brazilian context [47]. Seasonality, as measured by SARIMA models, is a critical factor in anticipating transmission peaks, as evidenced by the surveillance of hemorrhagic fever in China [48]. The application of regionalized modeling has revealed heterogeneous patterns of cryptococcosis mortality, with declining trends observed in the Southeast and stability recorded in the North, Northeast, and Central-West regions. These findings are indicative of disparities in health infrastructure and access to diagnostics. The stationarity of the series, achieved by differentiation and adjustment of parameters (*p*, d, q), is essential to capture temporal and seasonal correlations [49].

The integration of quantitative methods in the tracking of infectious diseases, as demonstrated by the development of the web-based dashboard with ARIMA modeling applied to cryptococcosis, offers a methodological framework that can be replicated for other diseases of global epidemiological relevance. In Brazil, initiatives such as the Alert-Early System for Outbreaks with Pandemic Potential, which integrates climate and epidemiological data into unified platforms, have the potential to optimize early warnings for infectious diseases. This integration indicates that this initiative is not merely a technological innovation, but rather a strategic necessity in a world grappling with the intertwined challenges of climate change and globalization [50].

The region-specific ARIMA projections enable three-tiered public health responses stratified by risk level. In high-mortality regions (Southeast/South), where forecasts indicate persistent burden, health authorities could prioritize the following: (a) annual pre-positioning of antifungal reserves at sentinel hospitals before peak transmission seasons, (b) integration of cryptococcal antigen screening into routine HIV viral load testing protocols, and (c) community alert systems activated when mortality trends deviate from forecasted confidence bands. For emerging-risk regions (North/Northeast), the gradually increasing trajectories suggest investments in diagnostic infrastructure—particularly point-of-care lumbar puncture kits in remote clinics—and telehealth training for disseminated cryptococcosis management. The utilization of region-specific ARIMA projections facilitates the implementation of targeted public health responses, which are stratified by risk level. To illustrate the operational implementation of this technology, consider the following hypothetical scenario. Using the dashboard’s interactive features, it is possible to identify anomalies and filter data in real-time to identify geographic groups and demographics. It also facilitates comprehension of the factors that influence the mobilization of resources.

Notwithstanding, the model developed establishes a replicable standard for analysis of other neglected diseases, with the potential for integration into national and international platforms. With access to the data, users can develop their applications and innovations in data visualization and perform new analyses aligned with their research objectives.

Future iterations will incorporate user engagement metrics to monitor adoption patterns and optimize feature prioritization, thereby identifying high-utilization components to guide iterative refinements. With Brazil’s national health registries, empirical evidence demonstrates that probabilistic linkage between SIM and Notifiable Diseases Information System databases significantly enriches epidemiological insights, particularly for HIV co-infection patterns [51]. Such integrations address fundamental limitations in mortality data by incorporating HIV serostatus, CD4 counts, and antiviral treatment histories—critical for cryptococcosis surveillance [52,53]. Furthermore, linkage with Brazilian Institute of Geography and Statistics census data enables population-normalized mortality rates, reducing ecological bias in regional comparisons [54].

Sustainability is operationally embedded through the CRISP-DM methodology’s cyclical phases, which facilitate continuous data pipeline optimization. Automated DATASUS API synchronization allows periodic incorporation of new mortality records, while modular design permits feature enhancements aligned with public health workflows [55,56]. Integration with Brazil’s Unified Health System digital ecosystem transforms the dashboard from a standalone tool into an operational surveillance node. Embedding within e-SUS Primary Care platforms enables bidirectional data exchange, leveraging existing infrastructure for Sustainability [53].

Open-source availability of dashboard code facilitates repurposing for digital health education. Health professionals can simulate epidemiological scenarios using live data streams, enhancing competency in outbreak response [57].

### Limitations

Exclusive reliance on the SIM introduces the possibility of reporting bias; incorporating complementary sources and record linkage techniques could mitigate this limitation. The absence of population-normalized rates in the main analyses stems from methodological constraints; SIM provides mortality data without integrated population denominators. Future iterations could incorporate Brazilian census data to generate regional incidence estimates, though retrospective standardization introduces additional assumptions.

While the SIM lacks HIV serostatus fields, integration with SINAN and SISCEL via probabilistic linkage (e.g., Reclink algorithms) enables retrospective co-infection analysis, as demonstrated in TB-HIV mortality studies achieving >90% match rates [7]. Geospatial imputation further identifies high-risk clusters where linkage is partial, supporting targeted interventions in regions with cryptococcosis incidence >3/100,000 inhabitants [9]. The integration of HIV serostatus data poses significant operational and regulatory challenges that necessitate meticulous consideration. The implementation of probabilistic linkage is contingent upon the utilization of stable matching variables, such as names and birthdates. However, research findings indicate that 18–22% of Brazilian HIV records are accompanied by incomplete demographic identifiers, a circumstance that has the potential to compromise the accuracy of matching processes. Regulatory frameworks present substantial barriers. Resolution CNS 738/2024 stipulates that ethics committee approval is necessary for the use of secondary data, thereby engendering delays of six to twelve months. Concurrently, Brazil’s General Data Protection Law mandates the establishment of intricate governance agreements for the dissemination of sensitive health data across systems [51,54].

## 5. Conclusions

This study developed an interactive and accessible data dashboard to monitor cryptococcosis-related deaths in Brazil between 2000 and 2022, addressing a critical gap in the epidemiologic surveillance of this systemic mycosis. The dashboard not only consolidates historical and sociodemographic data from the SIM but also provides a dynamic platform for visualizing, analyzing, and collecting these data, providing a technical foundation for the field of health information systems by demonstrating how interactive dashboards can transform raw data into actionable insights, especially for neglected diseases such as cryptococcosis. The methodological approach and results obtained provide a replicable model for other contexts, as long as they are adapted to the local specificities of health systems.

## Figures and Tables

**Figure 1 tropicalmed-10-00304-f001:**
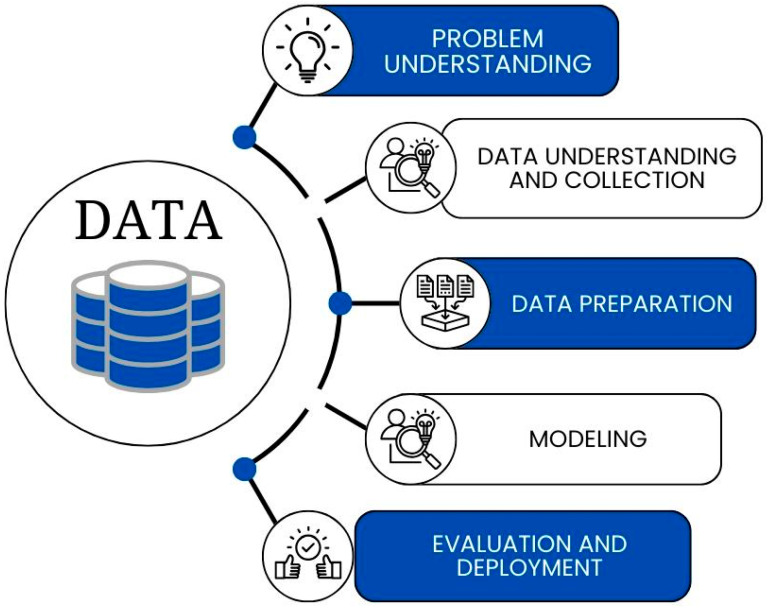
Iterative workflow of the Cross-Industry Standard Process for Data Mining methodology [20], formally comprising six interconnected phases: (1) business understanding (problem definition), (2) data understanding (initial exploration), (3) data preparation (cleaning/transformation), (4) modeling (algorithm development), (5) evaluation (validation), and (6) deployment (implementation). Bidirectional arrows denote cyclical refinement, where insights from later phases inform earlier stages.

**Figure 2 tropicalmed-10-00304-f002:**
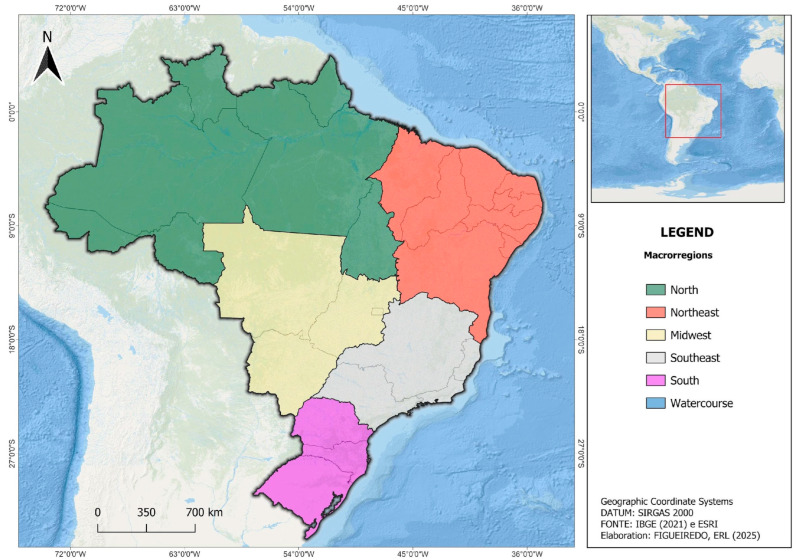
Political-administrative macroregions of Brazil: North, Northeast, Midwest, Southeast, and South [21].

**Figure 3 tropicalmed-10-00304-f003:**
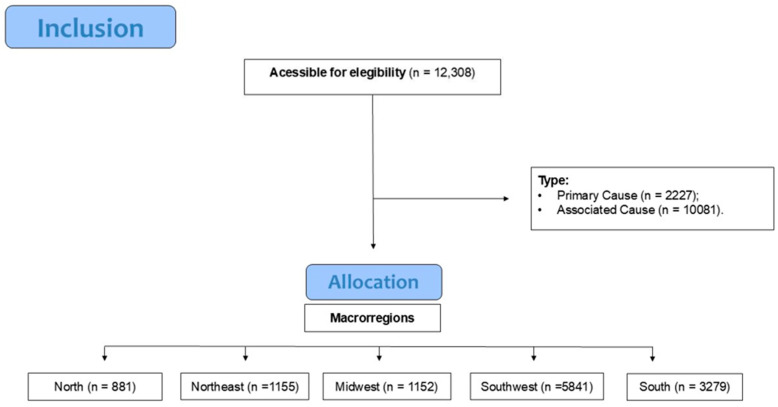
Data eligibility protocol for cryptococcosis-related mortality records (2000–2022). Inclusion criteria: ICD-10 codes B45-B459 in primary/associated causes and complete spatiotemporal variables (age, date, and municipality). Exclusion criteria: duplicate records, or missing core spatiotemporal variables, or implausible values (e.g., age > 120 years). Symbol ‘n’ denotes record counts at each stage. Final analytical sample: *n* = 12,308.

**Figure 4 tropicalmed-10-00304-f004:**
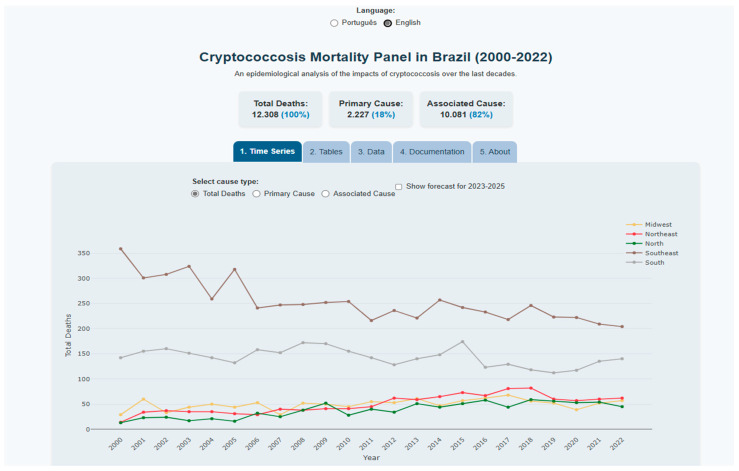
Dashboard time-series module displaying cryptococcosis mortality trends (2000–2022) stratified by Brazilian macroregion and cause type (primary/associated).

**Figure 5 tropicalmed-10-00304-f005:**
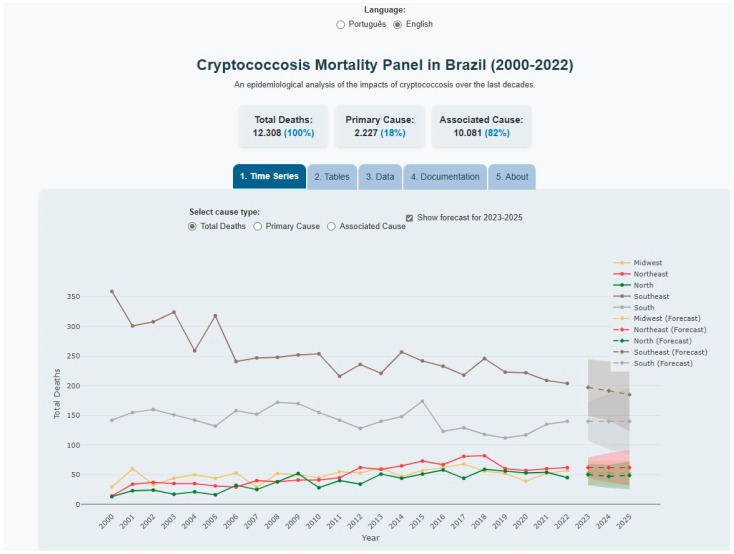
Regional ARIMA mortality projections (2023–2025) with historical data (solid lines), point forecasts (dashed lines), and 95% confidence intervals (shaded areas).

**Figure 6 tropicalmed-10-00304-f006:**
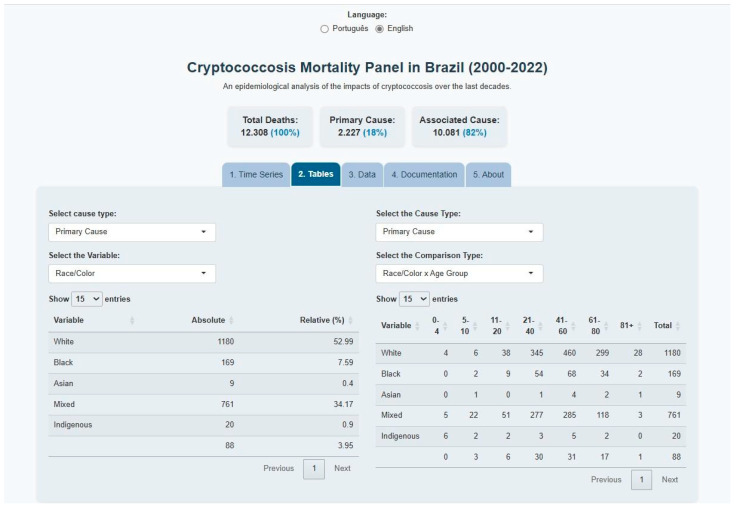
Interactive mortality frequency tables: distributions by race/ethnicity and age group for primary causes.

**Figure 7 tropicalmed-10-00304-f007:**
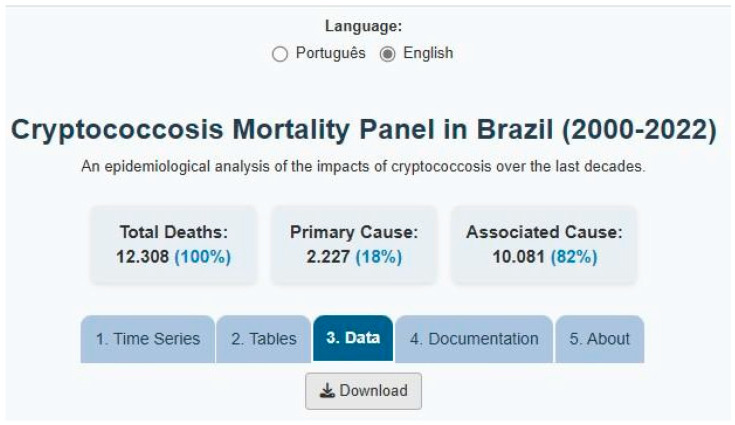
Data export interface providing downloadable mortality records (2000–2022) in open formats (XLSX/CSV).

## Data Availability

The data used for the study are publicly available and can be accessed from the website (opendatasus.saude.gov.br) and Microdatasus in RStudio (check the preprocessing script in the Appendix A).

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
