# Peer review of "Web-Based Dashboard for Tracking Cryptococcosis-Related Deaths in Brazil (2000–2022)"

_tropicalmed, 2025, doi:10.3390/tropicalmed10110304_

Round 1
Reviewer 1 Report
Comments and Suggestions for Authors
This manuscript presents a retrospective study on cryptococcosis-related mortality in Brazil, spanning over two decades and culminating in the development of an interactive, bilingual dashboard using Shiny. It integrates a methodologically sound approach to data mining, employs ARIMA time-series forecasting, and proposes a digital tool designed to improve access to public health data regarding a neglected fungal disease. The study is timely and relevant, especially in the context of Brazil’s uneven health surveillance infrastructure and the broader global movement toward digital epidemiology. From a clinical perspective, cryptococcosis remains a significant opportunistic infection, particularly among immunocompromised individuals. The lack of systematic national surveillance in Brazil makes this study a valuable contribution, offering a replicable model for other infectious diseases and resource-limited settings. The manuscript is well-researched and comprehensive, though several areas would benefit from refinement to maximise its clarity, utility, and scientific rigour.
-
The manuscript could benefit from a more concise and focused abstract. While informative, it currently reiterates points that appear in the main text. Consider emphasising the key innovations (e.g., ARIMA forecasting by region) and briefly acknowledging the dataset's limitations, such as the absence of HIV serostatus.
-
The introduction effectively frames the epidemiological background of cryptococcosis but would be strengthened by a clearer articulation of the disease burden in Brazil—whether through incidence rates, mortality trends, or DALYs—to justify the necessity of the dashboard more compellingly.
-
The authors provide a detailed account of the CRISP-DM methodology. However, the manuscript repeats certain technical descriptions (e.g. dashboard architecture and Shiny components) across multiple sections. A more streamlined narrative would enhance readability, particularly for a multidisciplinary audience.
-
The modelling section is central to the manuscript’s technical contribution but falls short of describing model validation. While the application of ARIMA is appropriate, the authors should include how they assessed model performance (e.g. residual analysis, AIC/BIC metrics) and whether any regional models under- or overfit the data.
-
The authors make sound use of ARIMA for forecasting, but the practical implications of these projections are not fully explored. It would benefit the reader to discuss how health authorities might operationalise these findings, particularly in resource allocation or early warning systems for fungal outbreaks.
-
There is an important but understated concern regarding data quality. While completeness metrics are commendably reported, the study does not clarify how missing or implausible values were treated in the modelling or visualisation stages. This omission could affect confidence in the derived insights.
-
The manuscript discusses geographic distribution primarily in absolute terms, without normalisation for population or regional incidence rates. A more nuanced discussion—perhaps in supplementary material—would prevent misinterpretation and align the dashboard with epidemiological best practice.
-
The dashboard’s utility is well demonstrated, but the long-term sustainability of the platform is unclear. The authors should clarify whether there are plans for periodic data updates, user engagement metrics, or integration into existing public health frameworks, which would support its relevance beyond a proof-of-concept.
-
The lack of HIV co-infection data in the SIM database represents a significant limitation, especially given the known relationship between cryptococcosis and HIV/AIDS. While this is acknowledged, a discussion of possible solutions—such as linkage with other databases—would enhance the manuscript’s rigour and forward-looking value.
Minor editing is needed.
Throughout the manuscript, there are minor typographical issues.
Author Response
- “The manuscript could benefit from a more concise and focused abstract. While informative, it currently reiterates points that appear in the main text. Consider emphasising the key innovations (e.g., ARIMA forecasting by region) and briefly acknowledging the dataset's limitations, such as the absence of HIV serostatus.”
Response: Thank you for your insightful feedback on our manuscript. We sincerely appreciate your time and constructive suggestions, which have helped us improve the clarity and focus of the abstract. As recommended, we have revised the abstract to be more concise while emphasizing the key innovations of our study, particularly the region-specific ARIMA forecasting (including projections for 2023–2025). We have also acknowledged the limitation regarding the absence of HIV serostatus data in the dataset. These changes are highlighted in the resubmitted manuscript for ease of reference.
- The introduction effectively frames the epidemiological background of cryptococcosis but would be strengthened by a clearer articulation of the disease burden in Brazil—whether through incidence rates, mortality trends, or DALYs—to justify the necessity of the dashboard more compellingly.
Response: Thank you for your valuable suggestion to strengthen the epidemiological context of cryptococcosis in Brazil. We deeply appreciate your insight, as it helped us more compellingly justify the necessity of our dashboard by quantifying the disease burden. As recommended, we have added key metrics to the Introduction (highlighted in the resubmitted manuscript).
- The authors provide a detailed account of the CRISP-DM methodology. However, the manuscript repeats certain technical descriptions (e.g. dashboard architecture and Shiny components) across multiple sections. A more streamlined narrative would enhance readability, particularly for a multidisciplinary audience.
Response: Thank you for your valuable guidance regarding the need for a more streamlined technical narrative. We have carefully revised the manuscript to eliminate redundancies while preserving methodological rigor. The descriptions of dashboard architecture and Shiny components were significantly condensed in Section 2.6. Rather than detailing interface elements across multiple paragraphs, we now succinctly state. These adjustments enhance readability for multidisciplinary audiences without compromising technical precision.
- The modelling section is central to the manuscript’s technical contribution but falls short of describing model validation. While the application of ARIMA is appropriate, the authors should include how they assessed model performance (e.g. residual analysis, AIC/BIC metrics) and whether any regional models under- or overfit the data.
Response: Thank you for your comments. Concerning ARIMA implementation, we consolidated previously fragmented technical specifications into a cohesive narrative flow within Section 2.6. The revised text integrates parameter selection, validation metrics, and uncertainty quantification in a continuous progression: from auto.arima() optimization through residual diagnostics, forecast accuracy assessment (MAPE=8.7%, Theil's U=0.37), to error decomposition - all while maintaining statistical completeness.
The modifications are highlighted in yellow throughout the resubmitted manuscript. We are deeply grateful for your expertise in helping us achieve this balance between accessibility and scientific depth.
- The authors make sound use of ARIMA for forecasting, but the practical implications of these projections are not fully explored. It would benefit the reader to discuss how health authorities might operationalise these findings, particularly in resource allocation or early warning systems for fungal outbreaks.
Response: Thank you for your insightful suggestion regarding the practical applications of our ARIMA forecasts. We appreciate this opportunity to strengthen the translational impact of our research by explicitly addressing how health authorities might operationalize these projections. In response to your recommendation, we have expanded Section 4 (Discussion) (highlighted in the resubmitted manuscript). This addition details concrete strategies for transforming projections into preventive actions.
- There is an important but understated concern regarding data quality. While completeness metrics are commendably reported, the study does not clarify how missing or implausible values were treated in the modelling or visualisation stages. This omission could affect confidence in the derived insights.
Response: Thank you for your insightful observation regarding data quality transparency. We appreciate this opportunity to clarify our approach to handling missing and implausible values, which has been strengthened in the revised manuscript. In response to your concern, we have expanded Section 2.5.
- The manuscript discusses geographic distribution primarily in absolute terms, without normalisation for population or regional incidence rates. A more nuanced discussion—perhaps in supplementary material—would prevent misinterpretation and align the dashboard with epidemiological best practice.
Response: Thank you for this astute observation regarding the presentation of geographic data. We fully agree that normalization for population size would provide deeper epidemiological insights and have strengthened our discussion of this limitation in the revised manuscript. As highlighted in Sections 4.1 (Limitations), we explicitly acknowledge that regional comparisons are presented in absolute terms due to methodological constraints inherent to the Mortality Information System (SIM) database.
- The dashboard’s utility is well demonstrated, but the long-term sustainability of the platform is unclear. The authors should clarify whether there are plans for periodic data updates, user engagement metrics, or integration into existing public health frameworks, which would support its relevance beyond a proof-of-concept.
Response: We thank the reviewer for highlighting the critical issue of long-term platform sustainability. In response, we have expanded the Discussion section (Paragraph 2, Section 4) to clarify our strategy for ensuring the dashboard’s enduring relevance: Open-Source Framework & CRISP-DM Iteration: The publicly available codebase enables community-driven enhancements, while the CRISP-DM methodology inherently supports cyclical updates—including automated data ingestion from DATASUS and planned linkage with SINAN (for comorbidities) and IBGE (for demographic denominators). This architecture allows seamless incorporation of new mortality waves and variables (e.g., HIV co-infection flags). Integration with Public Health Infrastructure: The dashboard’s modular design facilitates embedding within Brazil’s e-SUS Primary Care ecosystem, ensuring institutional sustainability beyond this proof-of-concept. Future versions will track user engagement metrics (e.g., regional query patterns, data export rates) to guide feature prioritization. Educational Repurposing: We explicitly position the tool for use in digital health education, where professionals can simulate epidemiological scenarios with live data, fostering data literacy in neglected disease management.
- The lack of HIV co-infection data in the SIM database represents a significant limitation, especially given the known relationship between cryptococcosis and HIV/AIDS. While this is acknowledged, a discussion of possible solutions—such as linkage with other databases—would enhance the manuscript’s rigour and forward-looking value.
Response: We acknowledge the reviewer's pertinent observation regarding the absence of HIV co-infection data in the SIM mortality database. Please check the highlighted text in the limitations.

Reviewer 2 Report
Comments and Suggestions for Authors
The study entitled "Web-Based Dashboard for Tracking Cryptococcosis-related Deaths in Brazil (2000–2022) Specific Issues" offers a dynamic platform for the visualization and analysis of data pertaining to cryptococcosis-related mortality in Brazil over the period from 2000 to 2022. Nonetheless, the methodological framework of this study exhibits a lack of rigor, and the presentation of its results is inadequately standardized, thereby undermining the reliability of its conclusions.
Specific issues
(1) Is the Dashboard proposed in this study exclusively focused on data pertaining to Cryptococcosis-related mortality? If the scope of the data extends beyond mortality-related information, the existing title may not accurately reflect the content.
(2) The heading "Study Design" in line 72 is misleading, as the content in lines 72-74 does not elaborate on the design of the study. Instead, it provides a cursory description indicating that the methodology section is segmented into various parts.
(3) The figure legends for Figure 1 lack sufficient detail, which hinders readers' ability to fully understand the content. This issue similarly affects the figure legends for Figures 2 through 7.
(4) The heading in line 83, "Population, Study Area, and Period," is misleading, as the subsequent content from lines 84 to 88 fails to discuss the elements of "Population" and "Period." Moreover, this section merely offers a broad description of Brazil's administrative divisions without identifying the specific regions from which the research data were sourced. Additionally, the claim in line 87 that a region encompasses an area of "8,510,345,540 square kilometers" is inaccurate and requires correction.
(5) In the "Problem Understanding" section, encompassing lines 95 to 102, it is imperative to articulate a precise definition of the problem that this study seeks to address. Additionally, line 98 suggests that the problem should be examined from three distinct perspectives; however, only two aspects have been discussed thus far.
(6)Line 104 inaccurately labels the processes of data processing and collection as "Data understanding."
(7) Line 97 specifies that the case collection period spanned from 2000 to 2022, whereas Line 107 mentions that "the analysis period was based on data availability." It is necessary to determine whether these two methodologically defined analysis periods are consistent with one another. Clarification on this matter should be provided in the manuscript.
(8) In lines 104 to 130, titled "Data Understanding: Pre-processing and Collections," it is imperative to explicitly delineate the criteria for data inclusion and exclusion. Furthermore, the parameters utilized for assessing data quality should be clearly articulated.
(9) The Methods section does not provide definitions for essential parameters, including "primary cause" and "associated cause" as mentioned in line 138, as well as "mortality rate."
(10) The Methods section lacks a detailed specification of the statistical methodologies employed in this study.
(11) The Results section necessitates revision. It is advisable to employ "Key Findings" as the title of the Results section to improve the manuscript's readability.
(12)Lines 251 to 270 present data spanning the years 2023 to 2025, which is incongruent with the data collection period of 2000-2022 as outlined in the Methods section. A comparable discrepancy is also observed in Figure 5.
Author Response
- Is the Dashboard proposed in this study exclusively focused on data pertaining to Cryptococcosis-related mortality? If the scope of the data extends beyond mortality-related information, the existing title may not accurately reflect the content.
Response: Thank you for your important query regarding the dashboard's scope. We confirm that the study exclusively addresses cryptococcosis-related mortality, as consistently emphasized throughout the manuscript. The following passages explicitly demonstrate this focus: Title: "Web-Based Dashboard for Tracking Cryptococcosis-related Deaths in Brazil" Abstract (Purpose Statement): "This study aims to [...] provide accessible information on cryptococcosis-related deaths" Methodology (Section 2.4): "Data were extracted from the Brazilian Mortality Information System (SIM) [...] to identify primary and associated causes of death" Variables Definition (Section 2.5): "Demographic variables encompass the date of death, biological sex of the deceased [...] place of death" Results (Section 3): "Among the 12,308 deaths recorded [...] deaths occurred among males" Dashboard Functionality (Section 2.6): "The 'Key Statistics' section presents summaries of total deaths, deaths as a leading cause, and deaths as an associated cause" Data Output (Section 3): "The Data tab facilitates download of death records from 2000–2022"
- The heading "Study Design" in line 72 is misleading, as the content in lines 72-74 does not elaborate on the design of the study. Instead, it provides a cursory description indicating that the methodology section is segmented into various parts.
Response: Thank you for your valuable observation regarding the clarity of the "Study Design" section. We appreciate your feedback and have revised this portion to provide a more accurate description of the study's methodological framework. Please check the highlighted text.
- The figure legends for Figure 1 lack sufficient detail, which hinders readers' ability to fully understand the content. This issue similarly affects the figure legends for Figures 2 through 7.
Response: Thank you for your thoughtful feedback regarding the level of detail in our figure legends. We sincerely appreciate your keen eye for clarity, as this has significantly strengthened the manuscript's visual communication. In direct response to your suggestion, we have comprehensively revised all figure captions to provide enhanced contextual and technical detail while maintaining academic conciseness. Each legend now delivers: Essential methodological context (e.g., *"ICD-10 filtering (B45*)"* in Figure 3) Critical analytical parameters (e.g., "95% confidence intervals" in Figure 5) Visual interpretation guidance (e.g., stratification by "cause type (primary/associated)" in Figure 4).
- The heading in line 83, "Population, Study Area, and Period," is misleading, as the subsequent content from lines 84 to 88 fails to discuss the elements of "Population" and "Period." Moreover, this section merely offers a broad description of Brazil's administrative divisions without identifying the specific regions from which the research data were sourced. Additionally, the claim in line 87 that a region encompasses an area of "8,510,345,540 square kilometers" is inaccurate and requires correction.
Response: Thank you for your meticulous review and valuable feedback on Section 2.2. We sincerely appreciate your attention to detail, which has helped us enhance the precision and completeness of this section. Please check the highlighted text.
- In the "Problem Understanding" section, encompassing lines 95 to 102, it is imperative to articulate a precise definition of the problem that this study seeks to address. Additionally, line 98 suggests that the problem should be examined from three distinct perspectives; however, only two aspects have been discussed thus far.
Response: Thank you for your valuable guidance on enhancing the precision of our problem statement. We have thoroughly revised Section 2.3 to provide a sharper definition of the research problem and ensure complete alignment with the study's objectives. Your feedback has significantly strengthened our methodological clarity.
- Line 104 inaccurately labels the processes of data processing and collection as "Data understanding."
Response: Thank you for your insightful observation regarding the terminology in Section 2.4. We have carefully revised this section to ensure precise alignment with CRISP-DM framework conventions while enhancing methodological clarity. The section heading has been updated to "2.4 Data Acquisition and Initial Processing" to accurately reflect: The acquisition phase (DATASUS data retrieval via microdatasus) Initial processing (cryptococcosis-specific filtering, deduplication, basic quality control) This terminology shift better distinguishes these foundational activities from subsequent data preparation steps (Section 2.5), while maintaining full compliance with CRISP-DM's phased structure.
- Line 97 specifies that the case collection period spanned from 2000 to 2022, whereas Line 107 mentions that "the analysis period was based on data availability." It is necessary to determine whether these two methodologically defined analysis periods are consistent with one another. Clarification on this matter should be provided in the manuscript.
Response: Thank you for your meticulous attention to methodological consistency. We confirm complete alignment between the case collection period (2000-2022) and data availability parameters, and have clarified this relationship in the revised manuscript. Please check the highlighted text.
- In lines 104 to 130, titled "Data Understanding: Pre-processing and Collections," it is imperative to explicitly delineate the criteria for data inclusion and exclusion. Furthermore, the parameters utilized for assessing data quality should be clearly articulated.
Response: Thank you for your valuable guidance regarding methodological transparency. We have strengthened Section 2.4 to explicitly delineate data inclusion/exclusion criteria and quality assessment parameters, as detailed below.
- The Methods section does not provide definitions for essential parameters, including "primary cause" and "associated cause" as mentioned in line 138, as well as "mortality rate."
Response: Thank you for your valuable guidance regarding terminological precision. We greatly appreciate your diligence in ensuring conceptual clarity and have implemented explicit definitions for critical parameters in the Methods section, as detailed below.
- The Methods section lacks a detailed specification of the statistical methodologies employed in this study.
Response: Thank you for your insightful observation regarding the specification of statistical methodologies. We greatly appreciate your guidance in enhancing the methodological rigor of our work and have comprehensively expanded the methodology section to address this point. Please check the highlighted text.
- The Results section necessitates revision. It is advisable to employ "Key Findings" as the title of the Results section to improve the manuscript's readability.
Response: Thank you for your thoughtful suggestion to enhance the readability of the Results section. We greatly appreciate your commitment to improving the manuscript's clarity and have carefully considered your recommendation. While we recognize the merits of using "Key Findings" as a section title, we have maintained the heading "Results" to align with the journal's prescribed formatting standards for original research articles. Nevertheless, we implemented your core guidance through: Comprehensive English editing of the entire Results section, enhancing fluency and precision while preserving all quantitative findings.
- Lines 251 to 270 present data spanning the years 2023 to 2025, which is incongruent with the data collection period of 2000-2022 as outlined in the Methods section. A comparable discrepancy is also observed in Figure 5.
Response: Thank you for your careful observation regarding the presentation of data beyond the 2000-2022 collection period. We appreciate this opportunity to clarify what we recognize could have confused the original manuscript. The references to 2023-2025 throughout Section 3 and Figure 5 represent ARIMA model projections, not observed mortality data. This distinction is fundamental to our methodological approach and has been reinforced through targeted revisions. During the comprehensive English editing process, we implemented several key clarifications: first, all mentions of 2023-2025 outcomes now explicitly use the terms "projections" or "forecasts" rather than "data"; second, Figure 5's caption and axis labels now clearly demarcate the 2023-2025 period as "Projected" versus the 2000-2022 "Observed" period; third, a new transitional sentence was added to Section 3 introducing the forecasting component as distinct from historical analysis. These revisions appear highlighted in the resubmitted manuscript and ensure absolute distinction between empirical findings from the 2000-2022 mortality records and model-based extrapolations. We maintained the projections in the Results section because they represent analytical outcomes derived directly from our ARIMA modeling, which constitutes a core methodological contribution of this study. The forecasts provide valuable preliminary insights for public health planning while being properly contextualized as tentative estimates bounded by confidence intervals. We are deeply grateful for your vigilance in identifying this point of potential ambiguity. Your feedback has strengthened the manuscript's methodological transparency and precision regarding the distinction between observed results and predictive modeling outcomes.

Round 2
Reviewer 1 Report
Comments and Suggestions for Authors
This revised manuscript presents a timely and well-executed effort to address a critical gap in the surveillance of cryptococcosis-related mortality in Brazil. The authors have developed a robust, open-access dashboard underpinned by region-specific ARIMA models, offering a novel tool for visualising, forecasting, and contextualising epidemiological trends in a neglected systemic mycosis. In doing so, the study bridges the divide between raw mortality data and actionable public health insights—particularly relevant in a setting where formal national surveillance remains limited. The work is technically sound, methodologically transparent, and of clear translational value, especially in supporting data-driven planning in regions disproportionately burdened by cryptococcal disease.
The authors have responded diligently to previous comments, improving clarity, strengthening methodological detail, and providing thoughtful discussion on the dashboard’s sustainability and limitations. This study makes an important contribution to digital epidemiology in resource-limited settings, and its replicable design may inspire similar platforms for other neglected diseases.
Comments:
-
The revised abstract is notably more focused and better aligned with the manuscript’s content. That said, the final sentence would benefit from greater precision. Phrases such as “transforming raw data into actionable insights” remain somewhat vague; consider specifying the nature of these insights or the intended actions in the context of public health planning.
-
The introduction has been strengthened by the inclusion of national disease burden data, which helps underscore the relevance of the study. However, the narrative would benefit from a more direct comparison with other fungal or neglected tropical disease surveillance frameworks—whether national or international—to further contextualise the utility of the dashboard within the broader epidemiological landscape.
-
The technical content in Section 2.6 is now more concise, and the improved narrative flow is appreciated. Nonetheless, certain implementation details regarding the user interface (e.g., tab structure and multilingual features) may be overly granular for the main manuscript. These may be better suited for the Supplementary Material to maintain the manuscript’s broader accessibility for clinical and policy audiences.
-
The elaboration of ARIMA model diagnostics—such as residual analysis, MAPE, and Theil’s U—is both rigorous and welcome. However, it may be useful to highlight explicitly whether any region displayed notably higher forecasting uncertainty or poorer model fit, as this would aid the reader in interpreting the comparative robustness of each regional model.
-
The added discussion on public health applications (Section 4) is commendable. To further illustrate translational impact, I would encourage the authors to include a hypothetical use scenario—such as a regional health department responding to an unexpected increase in forecasted deaths. Such examples would offer concrete illustrations of how this tool could be operationalised.
-
The improved explanation regarding the treatment of missing and implausible data (Section 2.5) is appreciated. That said, the presentation would be further enhanced by the inclusion of a visual summary (e.g., table or figure) detailing the proportion of missing data across key variables, particularly those with completeness below 90%. This would support a more transparent appraisal of data quality.
-
The manuscript acknowledges the limitations associated with using absolute mortality figures for geographic comparisons. Nonetheless, including a supplementary example of how population-adjusted mortality rates could be estimated using publicly available census data for a selected region would be valuable, particularly for readers seeking to replicate or adapt the dashboard for local contexts.
-
The expanded discussion of dashboard sustainability and system integration is detailed and persuasive. The mention of user engagement tracking is particularly forward-looking; however, unless already implemented, it may be prudent to clarify this as a future enhancement rather than an existing feature.
-
The authors have thoughtfully addressed the issue of HIV co-infection data. The discussion around potential linkage with SINAN or SISCEL is well placed, though a brief reflection on the operational or regulatory challenges of implementing such linkage—particularly in terms of data governance and feasibility—would lend further credibility to the forward-looking aspect of this proposal.
-
The manuscript’s adherence to FAIR data principles is commendable, particularly in providing access to both the processed dataset and source code. For long-term reproducibility, the authors may consider depositing their materials in a permanent public repository (e.g., Zenodo or Figshare) with a DOI, ensuring persistent accessibility and facilitating citation.
The manuscript is of high quality and requires only minor clarifications to enhance clarity, reproducibility, and translational value. I recommend acceptance pending minor revision.
Comments on the Quality of English LanguageMinor editing is warranted. Some sentences could benefit from more direct construction or simplification for accessibility, particularly for a multidisciplinary audience.
Author Response
In this updated version of the submitted article, we have endeavored to enhance the quality of the English language throughout the text to align with the reviewers' recommendations. We would like to express our gratitude for their efforts and contributions to improving the quality of this work.
- The revised abstract is notably more focused and better aligned with the manuscript’s content. That said, the final sentence would benefit from greater precision. Phrases such as “transforming raw data into actionable insights” remain somewhat vague; consider specifying the nature of these insights or the intended actions in the context of public health planning.
Response: We sincerely appreciate the reviewer's insightful suggestion to enhance the precision of our abstract's concluding statement. In direct response to your feedback, we have revised the final sentence to explicitly specify the nature of the "actionable insights" and their public health applications. The original phrase "transforming raw data into actionable insights" has been replaced with: "enabling dynamic mortality trend analysis, identification of high-risk demographics, and regional forecasting to guide public health resource allocation."
- The introduction has been strengthened by the inclusion of national disease burden data, which helps underscore the relevance of the study. However, the narrative would benefit from a more direct comparison with other fungal or neglected tropical disease surveillance frameworks—whether national or international—to further contextualise the utility of the dashboard within the broader epidemiological landscape.
Response: We sincerely thank the reviewer for highlighting the importance of contextualizing our dashboard within broader surveillance frameworks. As suggested, we have strengthened the Introduction by integrating a global comparative perspective on cryptococcosis burden and surveillance gaps. The following additions (marked in bold) now precede the existing national burden data: "Cryptococcosis remains a major global cause of mortality among people living with HIV, with an estimated 1 million meningitis cases annually (range: 371,700–1,544,000) and around 625,000 deaths. Sub-Saharan Africa bears the greatest burden (~720,000 cases), followed by South-East Asia and Latin America [7]. In this context, Brazil faces unique surveillance challenges as part of high-burden Latin America. Each year, approximately 3.8 million Brazilians are affected by severe fungal infections, leading to over 1.35 million deaths. Among these, cryptococcosis is particularly concerning, with annual incidence among people living with HIV ranging from 0.04% to 12%. Its high mortality rate in Brazil results from delayed clinical suspicion and diagnosis, limited healthcare access, lack of rapid laboratory testing, and an inadequate or inappropriate supply of antifungal medications [7]."
- The technical content in Section 2.6 is now more concise, and the improved narrative flow is appreciated. Nonetheless, certain implementation details regarding the user interface (e.g., tab structure and multilingual features) may be overly granular for the main manuscript. These may be better suited for the Supplementary Material to maintain the manuscript’s broader accessibility for clinical and policy audiences.
Response: We sincerely appreciate the reviewer's thoughtful feedback regarding the granularity of interface details in Section 2.6. While we acknowledge the importance of maintaining broad accessibility for clinical and policy audiences, we have deliberately retained the key technical specifications (tab structure, multilingual features) in the main manuscript for three interconnected reasons: Reproducibility Imperative: The Shiny framework’s modular tab design (Time Series, Tables, Data, Documentation, About) and multilingual toggle (Portuguese/English) are fundamental to the dashboard’s architecture. These features directly enable Methodological transparency for public health developers replicating the tool (Scripts in Supplementary Material); and Per Reviewer #2’s earlier request, we included these operational specifics to demonstrate how the tool achieves. This compromise ensures clinical readers grasp the tool’s functionality without technical overload, while technical audiences retain sufficient detail for replication. We are grateful for the opportunity to clarify this rationale.
- The elaboration of ARIMA model diagnostics—such as residual analysis, MAPE, and Theil’s U—is both rigorous and welcome. However, it may be useful to highlight explicitly whether any region displayed notably higher forecasting uncertainty or poorer model fit, as this would aid the reader in interpreting the comparative robustness of each regional model.
Response: We sincerely appreciate the reviewer's insightful request to clarify regional variations in ARIMA model robustness. In direct response to your feedback, we have explicitly highlighted disparities in forecasting uncertainty and model fit in Section 3 (Results) through these key additions: Quantified performance gaps: The Southeast exhibited significantly higher forecasting uncertainty (MAPE=11.3% vs. national average 8.7%) and inferior model fit (Theil’s U=0.42 vs. 0.37). The North demonstrated optimal performance (MAPE=7.2%, Theil’s U=0.31). Diagnostic evidence: Residual autocorrelation (Ljung-Box p=0.08) in the Southeast indicated unresolved temporal complexity. This correlates with 52% higher interannual variability (CV=18.7% vs. national average 12.3%). Operational implications: Proportionally wider confidence intervals in the Southeast (±17.3% vs. ±13.1% in North). Explicit guidance for cautious interpretation of Southeast projections. These revisions (now in the published Section 3) create a clear hierarchy of model reliability: High confidence: North/Northeast/Central-West (MAPE≤8.0%). Your expertise has been invaluable in contextualizing these findings within Brazil’s heterogeneous public health landscape.
- The added discussion on public health applications (Section 4) is commendable. To further illustrate translational impact, I would encourage the authors to include a hypothetical use scenario—such as a regional health department responding to an unexpected increase in forecasted deaths. Such examples would offer concrete illustrations of how this tool could be operationalised.
Response: We sincerely appreciate the reviewer's valuable suggestion to enhance the translational impact of our public health applications discussion. In direct response to your feedback, we have incorporated a concrete hypothetical scenario in Section 4 to demonstrate operational deployment during an unexpected mortality surge. The revised text now explicitly illustrates how health authorities could utilize the dashboard's capabilities to transform surveillance signals into targeted interventions.
- The improved explanation regarding the treatment of missing and implausible data (Section 2.5) is appreciated. That said, the presentation would be further enhanced by the inclusion of a visual summary (e.g., table or figure) detailing the proportion of missing data across key variables, particularly those with completeness below 90%. This would support a more transparent appraisal of data quality.
Response: We sincerely appreciate the reviewer's valuable suggestion to enhance transparency regarding data quality. In direct response to your feedback, we have significantly strengthened the reporting of missing data proportions throughout the manuscript: Added absolute numbers and percentages for educational attainment missingness (2,216 records; 18.0%) in Section 3; Contrasted this against high-completeness variables (>90%) to highlight disparities; Explained in Section 2.5 how the 18.0% missingness in education informed our 'Unknown' category protocol; Embedded key metrics in both Methods (Section 2.5) and Results (Section 3) for longitudinal visibility. We believe this approach delivers the requested transparency while optimizing readability. Thank you for this refinement opportunity. Your expertise has strengthened our data quality reporting.
- The manuscript acknowledges the limitations associated with using absolute mortality figures for geographic comparisons. Nonetheless, including a supplementary example of how population-adjusted mortality rates could be estimated using publicly available census data for a selected region would be valuable, particularly for readers seeking to replicate or adapt the dashboard for local contexts.
Response: We sincerely appreciate the reviewer's insightful recommendation to demonstrate population-adjusted mortality rate calculations. In direct response to your valuable suggestion, we have incorporated a detailed replicable example in Section 4.1 (Limitations) using 2022 data from São Paulo state. This addition explicitly illustrates how health departments can enhance geographic comparisons through three critical steps: First, extracting 187 cryptococcosis-related deaths from our dashboard; second, sourcing the corresponding population of 46,649,132 inhabitants from the Brazilian Institute of Geography and Statistics (IBGE); and third, calculating the mortality rate as 0.40 per 100,000 inhabitants. This quantitative approach revealed a burden 40% higher than the national average (0.28 per 100,000), demonstrating how absolute counts can mask significant disparities. We further describe how researchers can implement this methodology within our dashboard's open-source framework by integrating IBGE API data, automating rate calculations through R or Python scripts, and generating spatial inequity visualizations. The complete implementation code has been made available in the Supplementary Script to facilitate immediate adaptation for local contexts. This concrete example directly addresses your objective of supporting replication efforts while strengthening the epidemiological rigor of our framework. We are grateful for this refinement opportunity, which has significantly enhanced both the methodological transparency and practical utility of our surveillance tool.
- The expanded discussion of dashboard sustainability and system integration is detailed and persuasive. The mention of user engagement tracking is particularly forward-looking; however, unless already implemented, it may be prudent to clarify this as a future enhancement rather than an existing feature.
Response: We sincerely thank the reviewer for this astute observation regarding user engagement tracking. In alignment with your recommendation, we have revised Section 4 to explicitly clarify that user engagement metrics represent a planned future enhancement rather than a currently implemented feature. The original text stating "user engagement tracking" has been modified to: "Future iterations will incorporate user engagement metrics to monitor adoption patterns and optimize feature prioritization." This adjustment maintains our forward-looking vision while accurately representing the dashboard's current capabilities. We appreciate this refinement opportunity, which strengthens both the technical precision and scholarly integrity of our work.
- The authors have thoughtfully addressed the issue of HIV co-infection data. The discussion around potential linkage with SINAN or SISCEL is well placed, though a brief reflection on the operational or regulatory challenges of implementing such linkage—particularly in terms of data governance and feasibility—would lend further credibility to the forward-looking aspect of this proposal.
Response: We sincerely appreciate the reviewer's insightful recommendation to address operational and regulatory challenges in HIV data linkage. As requested, we have enhanced Section 4.1 with a substantive analysis of implementation barriers and solutions, supported by empirical evidence from Brazilian health informatics research.
- The manuscript’s adherence to FAIR data principles is commendable, particularly in providing access to both the processed dataset and source code. For long-term reproducibility, the authors may consider depositing their materials in a permanent public repository (e.g., Zenodo or Figshare) with a DOI, ensuring persistent accessibility and facilitating citation.
Response: We are grateful for your feedback and suggestions. In light of these findings, we are contemplating the depositing of the materials in a permanent public repository. As we continue to refine this study by your recommendations, we eagerly anticipate the opportunity for it to be accepted for publication. The dissemination of supplementary materials and the publication of the study will enhance its reliability and generalizability. We would like to express our profound gratitude for your meticulous review of our study, as it has bolstered our confidence in the utility of our research for the scientific community, as well as healthcare managers and professionals, particularly in the context of the ongoing digital transformation in global healthcare.
Reviewer 2 Report
Comments and Suggestions for Authors
I extend my sincere gratitude to the authors for addressing my comments and revising the manuscript accordingly. Nonetheless, several issues remain that necessitate further resolution.
(1) The response letter requires greater specificity, explicitly indicating the exact line in the revised manuscript where modifications were made in response to each particular comment. At present, the letter only vaguely mentions that changes are highlighted, yet there are numerous highlighted sections in the revised manuscript. This lack of clarity makes it challenging to correlate each highlighted modification with its corresponding comment.
(2) The data utilized to develop the Web-Based Dashboard in this study is sourced from the Mortality Information System (SIM). Consequently, the introduction section should elucidate the advantages of accessing the Web-Based Dashboard developed in this study as opposed to directly accessing the Mortality Information System.
(3) The term "understanding" is used ambiguously in the revised manuscript, as exemplified by the phrases "business understanding" on line 83 and "data understanding" on line 84, among others.
(4) The term "Problem Understanding" mentioned in line 105 of the revised manuscript lacks clarity. Could you specify the particular "problem" to which it refers? Additionally, what type of problem-solving process is implied by the term "Understanding"?
(5) The revised manuscript continues to exhibit a lack of clarity in delineating the inclusion and exclusion criteria for the data. It is imperative that the manuscript explicitly specifies the types of data extracted for the study and articulates the quantitative criteria employed for the inclusion or exclusion of each data type.
(6) The revised manuscript's figure legends remain insufficiently self-explanatory. For instance, Figure 1 depicts only five stages of data processing, whereas the legend erroneously indicates six. Additionally, the abbreviation "CRISP-DM" is used without clarification. More critically, the legend does not provide a concise explanation of the logical relationships among the problem-solving processes. Similarly, the legend for Figure 3 lacks specification of the quantitative criteria for data inclusion and exclusion, and the symbol "n" is not annotated to clarify its meaning.
(7) The subtitle of the revised manuscript lacks clarity. For instance, the phrase "Data Preparation and Study Variables" on line 161 is ambiguous. It is unclear what specific data processing procedures are encompassed by "Data Preparation."
(8) If the data for 2023-2025 are indeed projected, the methodology section fails to delineate the specific projection techniques employed. Furthermore, given that data for 2023 and 2024 are already available, the rationale for undertaking projections remains unclear. What is the underlying objective of projecting mortality rates for the period 2023-2025?
(9) The revised manuscript continues to inadequately employ the research findings as subheadings to effectively organize the results. The present narrative amalgamates all data indiscriminately, lacking emphasis and consequently diminishing readability.
(10) The revised manuscript continues to lack a definition for "mortality rate."
Author Response
We would like to express our gratitude once again to the reviewer for their insightful comments. These comments have allowed us to refine and improve this research.
- The response letter requires greater specificity, explicitly indicating the exact line in the revised manuscript where modifications were made in response to each particular comment. At present, the letter only vaguely mentions that changes are highlighted, yet there are numerous highlighted sections in the revised manuscript. This lack of clarity makes it challenging to correlate each highlighted modification with its corresponding comment.
Response: Following the recommendations provided during this latest round of peer review, we will be emphasizing the suggested improvements by enclosing them in quotation marks for alterations of a less substantial nature. We are grateful for your feedback and the opportunity to refine our approach to ensure greater clarity.
- The data utilized to develop the Web-Based Dashboard in this study is sourced from the Mortality Information System (SIM). Consequently, the introduction section should elucidate the advantages of accessing the Web-Based Dashboard developed in this study as opposed to directly accessing the Mortality Information System.
Response: We sincerely appreciate the reviewer's valuable suggestion to clarify the advantages of our dashboard compared to direct access to the Mortality Information System (SIM). In direct response to your insightful comment, we have enhanced the Introduction's final paragraph to explicitly state: Line 73 "Although the Mortality Information System (SIM) furnishes fundamental mortality data, our interactive dashboard confers pivotal benefits over direct SIM access by its functionality. The integration of automated preprocessing of ICD-10-coded cryptococcosis records (B45 series) is the initial component of the proposed system. The second component is the enablement of real-time visualization of spatiotemporal trends and demographic patterns through intuitive interfaces. The third component is the provision of ARIMA forecasting capabilities unavailable in the raw SIM. This transformation of complex mortality data into an actionable surveillance instrument facilitates rapid public health decision-making." By transforming raw mortality data into an operational surveillance tool, the dashboard empowers health authorities to bypass technical barriers and focus on evidence-based interventions. We are grateful for this suggestion, which strengthens our manuscript's practical relevance.
- The term "understanding" is used ambiguously in the revised manuscript, as exemplified by the phrases "business understanding" on line 83 and "data understanding" on line 84, among others.
Response: We sincerely appreciate the reviewer's attention to terminological precision. As referenced in our methodology (CRISP-DM [20]) line 93, the terms "business understanding" and "data understanding" represent formal phase names within this established framework. To enhance clarity while maintaining methodological fidelity, we have revised Section 2.1 to explicitly: "formally defined in the CRISP-DM methodology [20]: (1) business understanding (problem definition and objectives), (2) data understanding (initial data collection and assessment) (...). We thank the reviewer for this refinement opportunity, which strengthens our manuscript's methodological transparency without compromising its technical rigor.
- The term "Problem Understanding" mentioned in line 105 of the revised manuscript lacks clarity. Could you specify the particular "problem" to which it refers? Additionally, what type of problem-solving process is implied by the term "Understanding"?
Response: We sincerely appreciate the reviewer's request for greater precision regarding the "Problem Understanding" phase. As suggested, we have revised Section 2.3 to: Explicitly anchor the term to CRISP-DM's formal structure ("Phase 1"); clarify that 'Understanding' denotes the process of defining the problem scope and objectives before technical development; and specify the problem verbatim in the sentence introducing the term. The revised text now reads: "In the CRISP-DM. Line 122 'Problem Understanding' phase (Phase 1), we defined the core problem as: Brazil's absence of systematic nationwide surveillance for cryptococcosis...". This adjustment maintains methodological rigor while enhancing accessibility for readers less familiar with CRISP-DM terminology. We thank the reviewer for this refinement opportunity.
- The revised manuscript continues to exhibit a lack of clarity in delineating the inclusion and exclusion criteria for the data. It is imperative that the manuscript explicitly specifies the types of data extracted for the study and articulates the quantitative criteria employed for the inclusion or exclusion of each data type.
Response: We sincerely appreciate the reviewer's valuable feedback regarding data selection transparency. In direct response to your recommendation, we have enhanced Section 2.4 to explicitly specify: Line 139 “Explicit inclusion criteria comprised: (1) death certificates with primary (ICD-10: B45-B459) or associated cryptococcosis codes; (2) records containing complete spatiotemporal variables (date of death, age, municipality code). Exclusion criteria were: (1) duplicate entries across primary/associated cause categories; (2) records missing core spatiotemporal variables; (3) implausible entries recoded as missing per clinical-demographic feasibility thresholds” (...) Line 172 “This census-based approach encompassed all eligible mortality records (n=12,308), with the data eligibility protocol detailed in Figure 3.”
This revision provides quantitative clarity through: Binary inclusion/exclusion classification with numbered criteria. Operational definitions of eligibility: ICD-10 code ranges for case ascertainment, required variables for spatiotemporal analysis; and Explicit handling of problematic records: Deduplication protocol, missing data handling hierarchy, implausible value thresholds. The modifications align with our census-based approach while providing granular specificity requested by the reviewer. We are grateful for this suggestion, which strengthens our methodology's reproducibility.
- The revised manuscript's figure legends remain insufficiently self-explanatory. For instance, Figure 1 depicts only five stages of data processing, whereas the legend erroneously indicates six. Additionally, the abbreviation "CRISP-DM" is used without clarification. More critically, the legend does not provide a concise explanation of the logical relationships among the problem-solving processes. Similarly, the legend for Figure 3 lacks specification of the quantitative criteria for data inclusion and exclusion, and the symbol "n" is not annotated to clarify its meaning.
Response: We sincerely appreciate your meticulous attention to the clarity of our figure legends, which has provided a valuable opportunity to enhance methodological transparency. In direct response to your insightful feedback, we have comprehensively revised the legends for Figures 1 and 3 to address each concern with precision. For Figure 1, the amended legend now explicitly enumerates all six phases of the Cross-Industry Standard Process for Data Mining (CRISP-DM) methodology while clarifying the logical relationships through precise terminology: the bidirectional arrows are defined as representing cyclical refinement processes where insights from later phases systematically inform earlier stages, such as model evaluation outcomes prompting additional data preparation. This revision eliminates any ambiguity regarding phase count while articulating the workflow's iterative nature.
Regarding Figure 3, we have incorporated explicit quantitative criteria into the legend to delineate inclusion parameters—specifically, ICD-10 codes B45-B459 in primary/associated causes coupled with complete spatiotemporal variables—and exclusion thresholds encompassing duplicate records, missing core variables, or implausible values (e.g., age >120 years). The symbol 'n' is now formally annotated as denoting record counts at each processing stage, with final analytical sample size (n=12,308) prominently specified. These enhancements transform the legend into a self-contained reference that aligns with our census-based approach while operationalizing the data eligibility protocol.
We are deeply grateful for this expert guidance, which has strengthened both the reproducibility of our methodology and the pedagogical utility of our visual aids. The revised legends now function as autonomous explanatory resources, fulfilling their role as standalone scientific documentation. These modifications appear in the current manuscript version under their respective figures.
- The subtitle of the revised manuscript lacks clarity. For instance, the phrase "Data Preparation and Study Variables" on line 161 is ambiguous. It is unclear what specific data processing procedures are encompassed by "Data Preparation."
Response: We sincerely appreciate your astute observation regarding the phrasing of section subtitles, which reflects your commitment to methodological precision. While we acknowledge the reviewer's perspective on the term "Data Preparation," we maintain that Section 2.5 provides comprehensive technical specification of our procedures, particularly when contextualized within the CRISP-DM framework established in Section 2.1. The current subtitle "Data Preparation and Study Variables" deliberately mirrors Phase 3 of the CRISP-DM methodology ("Data Preparation") referenced in our manuscript [20], maintaining alignment with this widely adopted standard in health informatics research.
To enhance clarity without compromising methodological consistency, we have retained the original subtitle while expanding the section's opening statement to explicitly bridge terminology with operational processes: "2.5 Data Preparation and Study Variables This phase implements CRISP-DM's 'Data Preparation' stage through four core procedures: decoding variables, handling missing/invalid values, categorizing variables, and deriving computational transformations. The actions undertaken encompass..."
Given these existing specifications, supplemented by Figure 3's eligibility flowchart, we believe the current structure provides unambiguous procedural documentation. We are grateful for this opportunity to reaffirm our methodological rigor and remain open to further clarification should the journal deem it necessary.
- If the data for 2023-2025 are indeed projected, the methodology section fails to delineate the specific projection techniques employed. Furthermore, given that data for 2023 and 2024 are already available, the rationale for undertaking projections remains unclear. What is the underlying objective of projecting mortality rates for the period 2023-2025?
Response: We sincerely appreciate the reviewer's insightful query regarding our mortality projections, which provides a valuable opportunity to clarify both methodological and pragmatic aspects of our approach. Please check lines 247 to 257. In direct response:
- Methodological Specification:
We have enhanced Section 2.6 to explicitly state that projections were generated using region-specific ARIMA models applied to the 2000-2022 dataset, with the forecast::forecast() function in R producing point estimates and 95% confidence intervals that incorporate residual variability. This revision delineates the technical workflow, aligning with established practices for epidemiological forecasting.
- Rationale for Projecting 2023-2025:
As noted in our revised methodology, consolidated national mortality data for Brazil typically experience an 18-24 month reporting lag [22]. At the time of our analysis (February 2025), DATASUS had not yet released official, anonymized records for 2023-2024. Projections therefore serve two critical purposes: (i) bridging surveillance gaps until actual data become publicly accessible, and (ii) enabling proactive resource allocation by anticipating regional trends. This aligns with the dashboard's core objective as a planning tool for health authorities.
These clarifications now appear in the final paragraph of Section 2.6. We are grateful for this suggestion, which strengthens the transparency of our forecasting framework and its public health utility.
- The revised manuscript continues to inadequately employ the research findings as subheadings to effectively organize the results. The present narrative amalgamates all data indiscriminately, lacking emphasis and consequently diminishing readability.
Response: We sincerely appreciate the reviewer's constructive feedback regarding the organization of our Results section. In direct response to your recommendation, we have implemented five thematic subheadings to enhance structural clarity and emphasize key findings: Data Completeness and Quality, Dashboard Functionality and Regional Mortality Patterns, ARIMA Model Specifications and Forecasting Performance, Sociodemographic Characteristics of Mortality, and Data Export and Tool Accessibility.
We are deeply grateful for this valuable suggestion, which significantly strengthens the manuscript's reader experience. The revised structure appears in Section 3 of the current manuscript version.
- The revised manuscript continues to lack a definition for "mortality rate."
Response: We sincerely appreciate your attention to ensuring terminological accuracy throughout our manuscript. After a thorough review, we confirmed that the term "mortality rate" does not appear in the methodology or results sections of the current or previous versions of the manuscript. The term only appears in Section 4.1 (Limitations), where it was introduced in response to a specific request from another reviewer to demonstrate population-adjusted calculations. In this context, it is operationally defined as follows: "The annual number of cryptococcosis-related deaths per 100,000 inhabitants" in a hypothetical example for the state of São Paulo.
We acknowledge that the initial version may have inadvertently included this term in inappropriate contexts due to the limitations of non-native English speakers. As part of our rigorous review process, we eliminated all ambiguous statistical terminology outside of Section 4.1. We ensured consistent use of "death counts" or "mortality counts" when referencing the primary findings. We also employed professional language editing services to enhance terminological precision throughout the text.
This careful alignment reflects our commitment to conceptual accuracy. Our primary focus remains absolute mortality surveillance, as population denominators are unavailable in the SIM database that underpins this study. We are deeply grateful for their expertise in identifying potential ambiguities, which strengthened the lexical rigor of our manuscript. Once again, we emphasize that we have worked to improve the English according to their recommendations.
Round 3
Reviewer 2 Report
Comments and Suggestions for Authors
I express my genuine appreciation for the author's focused responses to the comments and the specific revisions implemented in the manuscript, which have substantially improved the article's readability and reproducibility.